# Prevalence of Dual Diagnoses among Children and Adolescents with Mental Health Conditions

**DOI:** 10.3390/children10020293

**Published:** 2023-02-02

**Authors:** Jandáč Tomáš, Šťastná Lenka

**Affiliations:** 1First Faculty of Medicine, Department of Addictology, Charles University, 12108 Prague, Czech Republic; 2Department of Addictology, General University Hospital in Prague, 12800 Prague, Czech Republic

**Keywords:** paedopsychiatry, addictology, dual diagnosis, children and adolescents

## Abstract

Background: The World Health Organisation defines dual diagnosis as the co-occurrence in the same individual of a psychoactive substance use disorder and another psychiatric disorder. Children and adolescents with dual diagnoses represent a significant public health burden in social and financial terms. Aims: The objective of the present paper is to provide a review of studies dealing with dual diagnoses and their prevalence among children and adolescents whose primary treatment involves psychiatric conditions. Methods: The PRISMA was used to conduct a systematic search. Articles published between January 2010 and May 2022 were searched for analysis. Results: Eight articles were eventually found eligible for the final content analysis. The analysis of the articles identified the prevalence of dual diagnoses among children and adolescents treated primarily for psychiatric conditions, the gender-specific occurrence of dual diagnoses, methods used to diagnose psychiatric and substance use disorders, types of psychiatric diagnoses involved in dual diagnoses, and prevalence differences contingent on the type of service provision as the main thematic areas. The prevalence of dual diagnoses among the target population ranged from 18.3% to 54% (mean 32.7%). Boys were more likely to experience dual diagnoses and affective disorders were the most frequent psychiatric diagnoses. Conclusion: The importance of the issue and the high prevalence of dual diagnoses make it imperative that this type of research is pursued.

## 1. Introduction

The World Health Organisation defines a dual diagnosis as the co-occurrence in the same individual of a psychoactive substance use disorder and another psychiatric disorder [1]. Dual diagnosis is a common term in addiction medicine, but it is often used inconsistently. The UK National Institute for Health and Care Excellence (NICE) defines dual diagnosis by referring to young people and adults with severe mental illness who misuse substances. Dual diagnosis appears to be a widespread term used on an everyday basis to refer to the concurrent presence of mental health issues and substance use, i.e., the situation where a person may or may not be formally diagnosed with a psychiatric disorder, dependence, or substance use disorder, or meets the formal criteria for such conditions [2]. Although the concept of dual diagnosis is widely accepted as the comorbidity of a psychiatric disorder and substance abuse, there has been growing interest from the field of disability in the meaning of dual diagnosis referred to as the coexistence of psychiatric disorders and intellectual disability [3]. Clinicians, researchers, and policy makers pay attention to this issue because of the challenges which are associated with the management of dual diagnoses, such as frequent relapses, poorer treatment engagement, and less satisfactory treatment outcomes. The higher risk of psychoactive substance use disorders is reported in the group of grown-up children with psychiatric disorders [4,5,6]. The issue of dual diagnoses among children and adolescents has received much less research interest, although dual diagnoses in this population represent a major social and financial burden for the public health and social welfare systems and are thus a pressing challenge for the national policy and public mental health [7]. Another reason why we decided to address the target population of children and adolescents is that the understanding of one of the factors for the development of substance use may be a step towards ameliorating the issue in paediatric practice. Severe health, social, and developmental consequences of substance use are major preventable symptoms among this target population [8]. The aim of the study was to present a review of papers dealing with the prevalence and other aspects of dual diagnoses among children and adolescents treated primarily for psychiatric conditions.

### 1.1. Prevalence

A study assessing the data of 1,036 participants aged 13-18 receiving mental health services found a 40.8% prevalence of substance use disorders [9]. A more recent U.S. study looking into the data of adolescents and young adults (aged 12–20) admitted to a psychiatric hospital primarily for mental health conditions found a 48% prevalence of dual diagnoses [10]. A British review [11] examining papers on dual diagnoses among patients with autism spectrum disorders focused on four areas, specifically epidemiology, patient characteristics, the function of drug use, and the effectiveness of treatment interventions. The review identified 18 papers as relevant, of which 11 concerned epidemiological studies. Only one, however, dealt with the prevalence of dual diagnoses in the population under scrutiny. The study with adolescents and young adults aged from 12 to 20 found no statistically significant gender-specific differences in the occurrence of dual diagnoses. Neither were there any differences identified in other demographic factors, such as cultural background and socioeconomic status. THC was the most commonly abused substance [10]. Another study systematically reviewed ten papers with the topic of dual diagnoses among adolescents. Reviewed studies were published from 1998 to 2004 and covered adolescents in both outpatient and inpatient care. The sample sizes of respondents varied from 89 to 992 individuals at a mean age of 16. The review suggested that inpatient services involve more severe cases of psychopathology and substance use related problems, which can occur in the case of outpatient care. The interviews used for the diagnostic of the participants included Diagnostic Interview Schedule for Children (DISC), the more concise Diagnostic Interview Schedule for Children-Predictive Scale, the Schedule for Affective Disorders and Schizophrenia for School-Age Children (K-SADS), and the Global Appraisal Individual Needs (GAIN). Some studies used only the assessment conducted by a psychiatrist. The levels of dual diagnoses ranged from 61% to 88%. The most commonly used substances included alcohol and THC. The prevalence of ADHD and conduct disorders was higher than that of anxiety disorders and mood disorders [7]. Another former study reported the prevalence of dual diagnosis in adolescents to be over 60% [5].

### 1.2. Types of Comorbidity

Mental disorders associated with dual diagnoses include anxiety disorders, depressive disorders, psychotic disorders, bipolar affective disorder, and antisocial personality disorder. While these associations were found among adult individuals, many of these mental disorders begin to develop during adolescence [12]. Some authors confirm that an increasing number of psychiatric disorders in one adolescent may elevate the risk of them developing substance use disorders. In bipolar adolescents, this risk was aggravated by the presence of oppositional defiant disorder and panic disorder, a family history of substance use, and low family cohesiveness [13]. In their review, Richardson [14] noted that risk factors for the development of dual diagnosis among patients with bipolar affective disorder included lower current age, lower age at the onset of symptoms, male gender, comorbid anxiety disorder, and attempted suicide. A four-year follow-up study of 627 lower secondary school students showed mental health disorders as predictors of the development of substance use disorders. Such mental health conditions included anxiety and depressive disorders [15]. In their study of dual diagnoses among adolescents, Wise et al. [5] reported depressive disorders being diagnosed in 24%, conduct disorders in 24%, ADHD in 11%, adjustment disorder in 7.7%, and bipolar affective disorder in 3.3% of the cases. Kim-Cohen [16] refers to the extensive literature on the dual diagnoses of conduct disorders and other mental health conditions among children and adolescents but points out the limited availability of evidence on comorbid substance use disorders despite the prevalence of conduct disorders being so high. The most commonly used substances were THC (91.2%), alcohol (60.8%), and cocaine (29.7%). Polydrug use was very frequent; it was found in 76.9% of the cases. The adolescents under study were clients of an inpatient residential facility. Disorders were diagnosed by a child psychiatrist using the DSM-IV criteria. The adolescents were aged 13–18, the mean age being 15.36 [5]. Some studies are focused on respondents who were hospitalised after suicidal behaviour but without a definite psychiatric disorder. People were hospitalized or attended services (outpatient and inpatient settings in regard to suicidal behaviour without specifying a psychiatric diagnosis). These studies were focused on the coexistence of suicidal behaviour and substance use disorders, or even overdose-related hospitalizations (substance use disorders) and suicidal behaviours in adolescents [17,18].

## 2. Materials and Methods

### 2.1. Eligibility Criteria

The search focused on journal articles dealing with the prevalence of dual diagnoses as their primary topic or secondary area of interest. Some studies refer to dual diagnosis as coexistence of psychiatric disorders and intellectual disability. We accept the meaning of dual diagnosis as coexistence of addictive disorders and psychiatric disorders. Determined selection criteria for the relevant publications were year of publication, key words, relevance of the article, type of publications, study design, language of the publications, and sociodemographic environment [19]. The search criteria were set as follows: (a) articles published between January 2010 and May 2022, (b) articles written in English, (c) texts published as peer-reviewed articles, chapters in scholarly books, and original studies, and (d) articles available in full text.

The articles were searched for in May 2022. The EBSCO, Medline, Scopus, and Web of Science databases, available to the researchers on the basis of their professional affiliation with the First Faculty of Medicine of Charles University, were used to conduct the search. Authors made an institutional section of databases. This collection of databases corresponds with recommendation for systematic reviews in the field of addictology [19]. The following terms were used as the key words: (juveniles OR adolescen* OR teen* OR child*) AND (psychiatric comorbidity OR dual diagnos?s OR concurrent disorder* OR psychopathology AND substance use OR alcohol use OR drug use OR addiction) AND prevalence. Two reviewers independently carried out study selection, evaluation, and data extraction.

### 2.2. Selection and Data Collection Process

The complete literature search process was recorded and documented. The PRISMA (Preferred Reporting Items for Systematic Reviews) protocol and guidelines were used to conduct the systematic review [20]. The initial search identified 571 studies; 27 items were removed as duplicate records (see the flow chart below, Figure 1). A total of 544 records were selected for further screening. In the next step, studies were excluded if their actual titles did not correspond with the target topic, i.e., mental health-specific dual diagnoses; and comorbidities concerned conditions other than psychiatric disorders and substance use disorders. 55 records were subjected to further and more thorough assessment, which resulted in 47 studies being excluded for failing to meet the selection criteria. One such criterion was the age of the study participants being above 19. Some studies returned by the search presented outcomes for children and adolescents, but they finally pooled their data with that of the general population sample and reported it in aggregate form. Other studies grouped children and adolescents with young adults. Nevertheless, the upper age limit for the young adult age category was inconsistent across studies. Another exclusion criterion applied to situations where the prevalence and aspects of a dual diagnosis were looked for among the target population of children and adolescents treated primarily for substance use disorders in drug services. Our search was aimed at the prevalence of dual diagnoses among children and adolescents treated primarily for mental health conditions in psychiatric hospitals and outpatient mental health facilities. The relevant study information was extracted into a Microsoft Excel database.

The following core areas of interest were identified within the final sample of the articles: (a) the prevalence of dual diagnoses among children and adolescents treated primarily for mental health conditions, (b) the gender-specific occurrence of dual diagnoses, (c) methods for diagnosing psychiatric and substance use disorders, (d) types of psychiatric diagnoses involved in dual diagnoses, and (e) prevalence variations depending on treatment modality.

### 2.3. Methods

We use descriptive statistics for the first main objective—the description of prevalence of dual diagnosis in the target group of children and adolescents with mental health conditions. We identify only eight studies with the information about the prevalence of dual diagnosis.

The content analysis focused on the areas of interest as outlined above. To ensure the continuity of the study, one researcher was appointed to screen the titles and abstracts and subsequently analyse the texts in order to identify the orientation of the topics. The results of the studies we had obtained are further summarized in a structured form as a table according to the classification criteria [19]. We used the Cochrane Collaboration’s tool for assessing risk of bias [21]. The following risks of bias were observed: (a) selection bias, (b) bias in measurements, (c) detection bias, (d) bias in incomplete outcome data, and (e) reporting bias. In terms of risk of bias in individual studies, most studies were assessed as having a high risk of bias in the categories selection bias, bias in measurements, and detection bias in studies Díaz et al., 2011; Hirschtritt et al., 2012; Masroor et al., 2019; Masroor et al., 2019; Wilens et al., 2013; Wu et al., 2011 [22,23,24,25,26]. Low risk was detected in reporting in the category of reporting bias. Regarding the overall quality of methodology, we assess the Norwegian study [27] conducted by Korsgaard et al. as high compared to the remaining studies.

### 2.4. Ethics

This paper involves no experimental research design studies with human or animal subjects.

## 3. Results

The final number of articles on the prevalence of dual diagnoses among children and adolescents in treatment for mental health conditions was eight. A summary is provided in Table 1.

### 3.1. Prevalence of Dual Diagnoses among Children and Adolescents Treated Primarily for Mental Health Conditions 

The prevalence of dual diagnoses among children and adolescents whose treatment primarily involved mental health conditions ranged from 18.3% to 54%. The difference between the lowest and highest values is thus 35.7 units. The mean is 32.7%, with the median being 28.3%. The prevalence levels of five studies [25,26,27,28,29] are summarised in Table 2.

Three studies [22,23,24] reported substance use disorders involving dual diagnoses broken down into substance-specific categories. These studies were not included in the calculation of the final prevalence of dual diagnoses.

However, we must mention the sample sizes of the studies are very different and the prevalence rates are not strictly comparable in terms of the understanding or determining the alleged relevance or value of the mean and median.

### 3.2. Gender-Specific Occurrence of Dual Diagnoses

Four studies [24,25,26,29] reported a higher risk of dual diagnoses among boys, with three of these quantifying the level of risk: one study noted dual diagnoses in 75% of the boys, another 54% of the boys, and the third study found the boys to be 1.6 times more likely than the girls to experience dual diagnoses. One study [22] found girls more likely to experience dual diagnoses, while another study [27] identified no significant gender-specific differences. Finally, two studies did not mention any gender-specific variations. 

### 3.3. Diagnostic Tools

The diagnostic tools used to determine psychiatric conditions and substance use disorders in dual diagnoses were described in the methodology sections of all the studies identified. Six studies referred to the use of the Diagnostic and Statistical Manual (DSM-IV) or assessment tools derived from it. Such tools included the K-SADS (Kiddie Schedule for Affective Disorders and Schizophrenia) referred to in two studies [23,25], M.I.N.I. (Mini-International Neuropsychiatric Interview), complementing the DSM-IV in one of the studies [27], YSR (Youth Self Report Form), used in addition to the DSM-IV in one of the studies [22], and the CBCL (Child Behavior Checklist) psychometric tool. Two studies reported the International Classification of Diseases (ICD-9) being used as one of the diagnostic tools.

The number of participants for whom the data had been collected was specified in all eight studies. The smallest sample comprised 34 individuals, while the largest one consisted of 800,614 respondents. A total of three studies involved analyses of large national data sets, while other studies examined facility-specific data. Five studies were from the United States; the remaining three originated in South Africa, Norway, and Spain.

### 3.4. Types of Psychiatric Diagnoses

The psychiatric diagnosis was described and discussed in all eight studies under scrutiny. The most common diagnoses were affective disorders, reported in six out of the eight studies. While conduct disorders were also mentioned in six out of the eight studies, their frequency was lower than that of affective disorders. In terms of frequency, anxiety and psychotic disorders came third and fourth, respectively, followed by ADHD. One study noted eating disorders among girls and one discussed obsessive-compulsive disorder.

### 3.5. Inpatient vs. Outpatient Care

Treatment modality was discussed in all of the studies under review. Wu et al. [26] found that patients with dual diagnoses in their examination of national health records were more likely to receive inpatient treatment. The studies indicating the highest [28] and the third highest [29] prevalence of dual diagnoses each reported inpatient participants. The studies referring to the second highest prevalence [25] and the lowest prevalence rates of dual diagnoses [22,27] reported outpatient care.

## 4. Discussion

Being aware of terminological inconsistency, we dealt with this issue by using alternative equivalents to “dual diagnosis” such as “psychiatric comorbidity”, “concurrent disorder”, and “concurrent psychopathology” in the early stages of the search. This approach was chosen because of the time frame of the studies under search, which was determined as spanning the period 2010–2021. During that time the terminology developed and the relevant terms were used inconsistently. The selection and initial screening of the texts showed that the search had returned articles dealing with the prevalence of dual diagnoses from two main perspectives. The first concerned the prevalence of dual diagnoses among children and adolescents with psychiatric disorders as their primary treatment issues. The other perspective involved the prevalence of dual diagnoses among children and adolescents with substance use disorders as their primary conditions. While in clinical terms it is the same situation, for the purposes of this systematic review we had to opt for one perspective only, specifically the former. The initial screening also revealed that the search had identified studies which dealt with the prevalence of dual diagnoses among children and adolescents, but the upper age limit of the subjects varied significantly at the researchers’ discretion. The most common option chosen by the researchers was aggregation with the population of young adults. The greatest range, specifically 11–35 years, was adopted by Mexican researchers [12]. In addition to reducing the number of valid studies identified for our work, this inconsistent definition of the “children and adolescents” group makes it more difficult to provide an accurate description of the specific target population. We, therefore, suggest a specific definition of the population in further research. 

The final number of studies included in our review was eight. This small number corresponded with the conclusions of a review addressing dual diagnoses in patients with autism spectrum disorders [11], where only 1 out of 11 epidemiological studies dealt with the prevalence of dual diagnoses. Another review [7] identified ten relevant articles. Its authors noted that while the issue of dual diagnoses was covered extensively for the adult population, the research into this topic in relation to children and adolescents was limited. Our results support this argument and indicate that it is advisable to conduct more epidemiological studies that look into dual diagnoses among children and adolescents.

The mean prevalence of dual diagnoses among children and adolescents treated primarily for mental health conditions was 32.7%. This result does not reach the levels reported by Aarons et al. [9] and Stephens et al. [10], who reported prevalence rates of 40.8% and 48%, respectively. It must be noted, however, that the latter study involved an extended target population (12–20 years old). Although our final prevalence rate does not reflect the levels reported by the above studies, it still indicates a high percentage of dual diagnoses in the target population. The difference between the results may be due to the fact that not all substance users fulfil the diagnostic criteria for a dependence syndrome. Accordingly, the above studies may have not taken account of any instances of harmful use in their evaluation of dual diagnoses. 

The majority of the studies identified being of male gender as a risk factor for dual diagnoses. This finding corresponded with the conclusions of Richardson [14], who studied substance use disorders among participants with bipolar affective disorder. On the other hand, this result does not match the findings reported by Stephens et al. [10], who identified no statistically significant gender-specific differences. The prevalence of dual diagnoses was higher among hospitalised patients.

The most common tool used to diagnose psychiatric and substance use disorders in the studies under review was the Diagnostic and Statistical Manual (DSM-IV) and measures derived from it. All the studies under scrutiny described the diagnostic tools used. We believe that it is useful to provide thorough descriptions of the assessment tools applied and recommend that such specifications should be encouraged in further studies dealing with the prevalence of dual diagnoses among children and adolescents treated primarily for substance use disorders.

In addition to substance disorders, affective disorders and conduct disorders were the most common psychiatric conditions diagnosed. The high rates of affective disorders and conduct disorders correspond with the conclusions of Wise et al. [5] and Wolitzky-Taylor et al. [15]. As our review affirms the high levels of comorbid substance use disorders and conduct disorders in dual diagnoses, we support the conclusions drawn by Kim-Cohen et al. [16] to the effect that the relationship between substance use disorders and conduct disorders should be investigated further. In more general terms, we recommend that any associations between the types of psychiatric comorbidities and substance-specific use disorders in dual diagnoses are analysed. Our research has identified no relationships across the studies under scrutiny, as such relationships were not addressed by this research.

The strength of this review is in the selection of a target group of children and adolescents. Limitations result from the fact that we included texts written only in English and we had limited access to databases; we suggest wider access to databases in the next study. Some studies had the prevalence of dual diagnosis as a partial topic. Additionally, studies in our review used different measurements and had different objectives and we recognised a small number of relevant studies; therefore, this study is more descriptive than comparative. This review focus on the studies that include patients with established diagnoses of psychiatric disorders and their comorbidity with substance use disorders in dual diagnoses. There are studies that focus on the coexistence of suicidal behaviour and substance use disorders or even overdose-related hospitalizations and suicidal behaviours in adolescents without a definite psychiatric disorder. We suggest accepting these studies in further reviews about the topic of dual diagnosis.

## 5. Conclusions

Our findings showed that the literature on dual diagnoses in the target group of children and adolescents with mental health conditions is scarce and this is one of few reviews in this topic. Better knowledge of types of psychiatric disorders in dual diagnosis can help in the care provided and clinicians can propose appropriate interventions. It is necessary to focus on the core of psychiatric disorders in adolescents to prevent substance use disorders. It seems the best way is to treat in multidisciplinary teams and focus on psychiatric disorders and substance use disorders at the same time. We could also struggle with the stigma of the target group if we know more about the aetiology of these disorders.

## Figures and Tables

**Figure 1 children-10-00293-f001:**
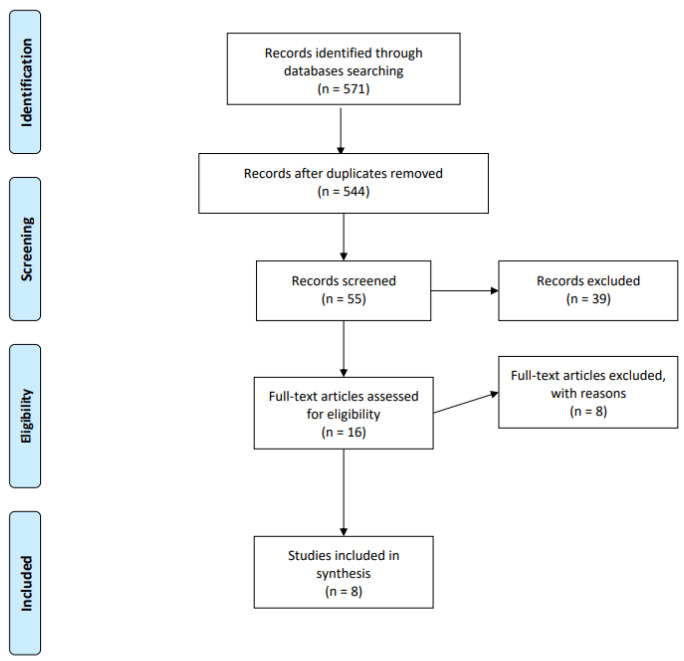
Prisma flow diagram.

**Table 1 children-10-00293-t001:** Papers included in the study.

Study (First Author)	Year of Publication	Country	Sample (n = x)	Range in Sample (Years)	Prevalence of Dual Diagnosis (%)	Interview PD and SUD
Diáz et al., 2011 [22]	2011	Spain	237	12–17	THC 10.1% alcohol 3.4% others 0.4%	DSM IV (YSR and CBCL)
Wu et al., 2011 [26]	2011	USA	n_1_ = 6210 n_2_ = 5247	2–12 13–17	1.6% 25%	DSM IV
Hirschtritt et al., 2012 [23]	2012	USA	34	12–17	THC abuse 47% (41.2% dependence) alcohol 29.4% (8.8%) polyvalent 2.9%	DSM IV KSADS
Lachman et al., 2012 [28]	2012	RSA	141	13–18	54%	DSM IV
Wilens et al., 2013 [25]	2013	USA	303	10–18	BD subjects 30% CD subjects 42%	DSM IV KSADS-E SCID
Hollen & Oritz, 2015 [29]	2015	USA	9154	11–17	25%	ICD 9
Korsgaard et al., 2016 [27]	2016	Norway	153	14–17	18.3 %	DSM IV M.I.N.I.
Masroor et al., 2019 [24]	2019	USA	800614	12–18	Amphetamins70.8% opioids 66.7% alcohol 52.7% THC 50.9%	ICD 9

Note: BD: bipolar disorder, CD: conduct disorder, PD: Psychiatric disorders, and SUD: substance use disorder.

**Table 2 children-10-00293-t002:** Prevalence of dual diagnosis.

Prevalence of Dual Diagnosis (%)	
mean	33.9
median	30
st.dev.	4.7
max.	49.1
min	18.3
range	30.8

Note: The prevalence of five studies [25,26,27,28,29].

## Data Availability

Original papers included in systematic review could be found in the EBSCO, Medline, Scopus, and Web of Science databases.

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
