# Peer review of "Prevalence of Dual Diagnoses among Children and Adolescents with Mental Health Conditions"

_children, 2023, doi:10.3390/children10020293_

Round 1

Reviewer 1 Report

The topic of the paper is interesting because there are few data about dual diagnoses in children and adolescents. In this regard, perhaps the paper lack of more extensive data on the background of this topic prior to 2010 (the authors performed the systematic search between January 2010 and May 2022). The Introduction only cites one paper specifically related to this issue (Couwenbergh et al) without explaining the results or other relevant data found so far or until the beginning of the authors’ search (articles such as those from Brian et al: Dual diagnosis and successful participation of adolescents in substance abuse treatment; or Biederman et al: Is ADHD a risk factor for psychoactive substance use disorders: Findings from a four-year prospective follow-up study).

Although the concept of dual diagnosis is widely accepted as the comorbidity of psychiatric disorder and substance abuse, there has been growing interest from the field of disability in the meaning of dual diagnosis referred to the coexistence of psychiatric disorders and intellectual disability (see, for instance, the work of Jeanne Farr, National Association of Dual Diagnosis). Perhaps some words clarifying the distinction of the two concepts could be added when referring to the selection of the studies reviewed.

In the “Materials and Methods” section, I find some confusing the expression “Determined selection criteria for the relevant publications were selected according to the classification that follow year of publication…”. In the same section, the acronym STROBE lacks reference or description.

Redundant expressions or sentences: “For assessing risk of bias was used the Cochrane Collaboration´s tool for assessing risk of bias [24].”

Expressions or sentences that need improvement: “In terms of risk of bias in individual studies most studies were assessed as the high risk of bias in categories selection bias, bias in measurements and detection bias [15, 16, 17, 19, 20]. Regarding the overall quality of methodology, we assess Norwegian study [18] conducted by Korsgaard et al. as high compared to the remaining studies.”

The numbers of the references in the text must be reviewed, as, for example, the study of Korsgaard (“Regarding the overall quality of methodology, we assess Norwegian study [18] conducted by Korsgaard et al. as high compared to the remaining studies.”) is not referenced as 18 but as 21 in de References section.

Results: even it become obvious/predictable, maybe it should be better to specify the meaning of the acronyms: “Interview PD (psychiatric disorders) and SUD (substance use disorders)”. Also in table 1: “BD subjects 30 % CD subjects 42 %”.

“3.1.1. Prevalence of dual diagnoses among children and adolescents treated primarily for mental health conditions”: As the sample sizes of the studies are so different, and after three studies being excluded, probably the prevalence rates are not strictly comparable, in terms of the understanding or determining the alleged relevance or value of the mean, median, SD… of the five studies finally included in this comparison.

Table 2 is perhaps unnecessary or must be completed with data such as which studies (5) are included, in spite of being explained in the text.

Diagnostic tools: “Diagnostic and Statistical Manual (DSM-IV) or assessment tools derived from it.” All studies specified the use of diagnostic interviews or simply based on DSM criteria?

To be considered: Patients who are hospitalised after suicidal behaviour but without a definite psychiatric disorder. This review focus on the studies that includes patients with established diagnoses of psychiatric disorders and their comorbidity with substance use disorders, but there are some cases which are hospitalized or attended in emergency rooms (or even in outpatient settings) who are labelled as “suicidal behaviour” without specifying a psychiatric diagnosis. In fact, there are studies that focus on the coexistence of suicidal behaviour and substance use disorders, or even overdose-related hospitalizations (Substance Use Disorders (SUD) and Suicidal Behaviors in Adolescents: Insights From Cross-Sectional Inpatient Study. Sayeda A Basith, Miles M Nakaska , Albulena Sejdiu, Aabha Shakya, Vaishalee Namdev, Siddharth Gupta, Keerthika Mathialagan , Ramkrishna Makani; Heroin Overdose-Related Child and Adolescent Hospitalizations: Insight on Comorbid Psychiatric and Substance Use Disorders. Uwandu Queeneth, Narmada N Bhimanadham, Pranita Mainali, Henry K Onyeaka, Amaya Pankaj, Rikinkumar S Patel). Perhaps this area should deserve some comments or consideration.

The few studies that finally met the author’s criteria and the heterogeneity of their samples, regions, ages ranges, type of care/services provision…  perhaps undermine the interest and generalizability of the paper, and probably compel to explain more deeply their characteristics, but this issue is partially solved in the Discussion section. Maybe a more detailed description of the differences among the studies would improve the paper, although the authors explain properly that “this study is more descriptive than comparative”.

Author Response

Response to Reviewer 1 Comments

Point 1: The Introduction only cites one paper specifically related to this issue (Couwenbergh et al) without explaining the results or other relevant data found so far or until the beginning of the authors’ search (articles such as those from Brian et al: Dual diagnosis and successful participation of adolescents in substance abuse treatment; or Biederman et al: Is ADHD a risk factor for psychoactive substance use disorders: Findings from a four-year prospective follow-up study).

Response 1: We added recommended studies in the introduction. (in red)

Point 2: Although the concept of dual diagnosis is widely accepted as the comorbidity of psychiatric disorder and substance abuse, there has been growing interest from the field of disability in the meaning of dual diagnosis referred to the coexistence of psychiatric disorders and intellectual disability (see, for instance, the work of Jeanne Farr, National Association of Dual Diagnosis). Perhaps some words clarifying the distinction of the two concepts could be added when referring to the selection of the studies reviewed.

Response 2: We meant it in the part of eligibility criteria. (in red)

Point 3: In the “Materials and Methods” section, I find some confusing the expression “Determined selection criteria for the relevant publications were selected according to the classification that follow year of publication…”. In the same section, the acronym STROBE lacks reference or description. Redundant expressions or sentences: “For assessing risk of bias was used the Cochrane Collaboration´s tool for assessing risk of bias [24].”

Response 3: It seems it was lost in translation. We revised the sentences in the better condition (in red)

Point 4: Expressions or sentences that need improvement: “In terms of risk of bias in individual studies most studies were assessed as the high risk of bias in categories selection bias, bias in measurements and detection bias [15, 16, 17, 19, 20]. Regarding the overall quality of methodology, we assess Norwegian study [18] conducted by Korsgaard et al. as high compared to the remaining studies.”

The numbers of the references in the text must be reviewed, as, for example, the study of Korsgaard (“Regarding the overall quality of methodology, we assess Norwegian study [18] conducted by Korsgaard et al. as high compared to the remaining studies.”) is not referenced as 18 but as 21 in de References section.

Response 4: We improved the sentences and checked citations. (in red)

Point 5: Results: even it become obvious/predictable, maybe it should be better to specify the meaning of the acronyms: “Interview PD (psychiatric disorders) and SUD (substance use disorders)”. Also in table 1: “BD subjects 30 % CD subjects 42 %”.

Response 5: We added “Note” bellow the Table 1.

Point 6: “3.1.1. Prevalence of dual diagnoses among children and adolescents treated primarily for mental health conditions”: As the sample sizes of the studies are so different, and after three studies being excluded, probably the prevalence rates are not strictly comparable, in terms of the understanding or determining the alleged relevance or value of the mean, median, SD… of the five studies finally included in this comparison

Response 6: We added the mention of explanation this limitation of understanding of the mean and median. (in red)

Point 7: Table 2 is perhaps unnecessary or must be completed with data such as which studies (5) are included, in spite of being explained in the text.

Response 7: We added the explanation in text and in the “Note” under the table. (in red)

Point 8: Diagnostic tools: “Diagnostic and Statistical Manual (DSM-IV) or assessment tools derived from it.” All studies specified the use of diagnostic interviews or simply based on DSM criteria?

Response 8: Six studies referred to the use of the Diagnostic and Statistical Manual (DSM-IV) or assessment tools derived from it. Two studies reported the International Classification of Diseases (ICD-9) being used as one of the diagnostic tools.

Point 9: To be considered: Patients who are hospitalised after suicidal behaviour but without a definite psychiatric disorder. This review focus on the studies that includes patients with established diagnoses of psychiatric disorders and their comorbidity with substance use disorders, but there are some cases which are hospitalized or attended in emergency rooms (or even in outpatient settings) who are labelled as “suicidal behaviour” without specifying a psychiatric diagnosis. In fact, there are studies that focus on the coexistence of suicidal behaviour and substance use disorders, or even overdose-related hospitalizations (Substance Use Disorders (SUD) and Suicidal Behaviors in Adolescents: Insights From Cross-Sectional Inpatient Study. Sayeda A Basith, Miles M Nakaska , Albulena Sejdiu, Aabha Shakya, Vaishalee Namdev, Siddharth Gupta, Keerthika Mathialagan , Ramkrishna Makani; Heroin Overdose-Related Child and Adolescent Hospitalizations: Insight on Comorbid Psychiatric and Substance Use Disorders. Uwandu Queeneth, Narmada N Bhimanadham, Pranita Mainali, Henry K Onyeaka, Amaya Pankaj, Rikinkumar S Patel). Perhaps this area should deserve some comments or consideration.

Response 9: It is great point. We discuss it in the part “Discussion” with our suggestion. (in red)

Point 10: The few studies that finally met the author’s criteria and the heterogeneity of their samples, regions, ages ranges, type of care/services provision…  perhaps undermine the interest and generalizability of the paper, and probably compel to explain more deeply their characteristics, but this issue is partially solved in the Discussion section. Maybe a more detailed description of the differences among the studies would improve the paper, although the authors explain properly that “this study is more descriptive than comparative”.

Response 10: I think this is not revision, but yes, we fulfill this point in the part of discussion and it is the recommendation for our further work.

Reviewer 2 Report

The introduction provide insufficient background information. Please, be include more relevant reference and information.  

Author Response

Response to Reviewer 2 Comments

Point 1: The introduction provide insufficient background information. Please, be include more relevant reference and information.  

Response 1: We improve the part of introduction for more relevant references and information- (in green)

Reviewer 3 Report

Dear Editor and Authors,

The article entitled as „Prevalence of Dual Diagnoses among Children and Adolescents with Mental Health Conditions” represents a systematic review article with an aim to provide summation of relevant studies dealt with dual diagnosis and its prevalence among children and adolescents whose primary treatment involves psychiatric conditions. As the mental health of younger population is often vulnerable nowadays, the presents of dual diagnosis among them is also very common. While growing, most of the diagnoses are present in the adulthood too and they influence life on many different personal levels  but at the end, thy can make challenges in the health investments looking at the government level, so this article is of the great importance and its subject suits to the Children.

I am very pleased to contribute with the review of this article and hope that my comments will help authors to improve their work.

I suggest authors to consider following suggestions:

Abstract – informative and very well presented.

Introduction – not informative enough, I suggest extending entire Introduction and adding the aims of the study at the end of Introduction as a separate paragraph. Using the newest articles on this subject to enrich the References and provide the better explanation of the complex problems that children and adolescence are dealing with, is necessary. Also, I suggest adding the view from the medical perspective in the sense of quality of healthcare that these individuals receive. Adding the comorbid-diseases that may follow psychiatric disorders and how psychiatric disorders influence the global burden of diseases, in few sentences, would improve the quality of the Introduction.

Methodology - Clearly defined. I suggest adding a PRISMA flow diagram. The form of DRISMA diagram can be found here: https://prisma-statement.org/prismastatement/flowdiagram.aspx?AspxAutoDetectCookieSupport=1

Discussion: Line 163 should be placed at the end of the Introduction and should be removed from this section.  Line 230 should be rewritten, especially the part with statement “worth noting”. Overall the Discussion is well-written.

Conclusion: Line 237 ..“and to the best of our knowledge”..  should be removed. Describe in few sentences more about importance of mental health prevention in these age groups, and note the importance of the role of medical practitioners in education of these individuals.

References: There is a double numeration of the reference list and it should be omitted. References are relevant and related to the subject but I would suggest adding more recent publications.

Sincerely,

Author Response

Response to Reviewer 3 Comments

Point 1: Introduction – not informative enough, I suggest extending entire Introduction and adding the aims of the study at the end of Introduction as a separate paragraph. Using the newest articles on this subject to enrich the References and provide the better explanation of the complex problems that children and adolescence are dealing with, is necessary. Also, I suggest adding the view from the medical perspective in the sense of quality of healthcare that these individuals receive. Adding the comorbid-diseases that may follow psychiatric disorders and how psychiatric disorders influence the global burden of diseases, in few sentences, would improve the quality of the Introduction.

Response 1: We improved the part of introduction. We enhance this part with the content of more relevant studies (in green and red)

Point 2: Methodology - Clearly defined. I suggest adding a PRISMA flow diagram. The form of PRISMA diagram can be found here: https://prisma-statement.org/prismastatement/flowdiagram.aspx?AspxAutoDetectCookieSupport=1

Response 2: We sent this diagram to editorial board earlier.

Point 3: Discussion: Line 163 should be placed at the end of the Introduction and should be removed from this section.  Line 230 should be rewritten, especially the part with statement “worth noting”. Overall the Discussion is well-written.

Response 3: The line 163 was replaced to the intruduction. The line 230 and overall part of limitations were rewritten. (in blue and red).

Point 4: Conclusion: Line 237 ..“and to the best of our knowledge”..  should be removed. Describe in few sentences more about importance of mental health prevention in these age groups, and note the importance of the role of medical practitioners in education of these individuals.

Response 4: The line 237 and its expression „and to the best of our knowledge“ was removed. We added few sentences in conclusion. (in blue)

Point 5: References: There is a double numeration of the reference list and it should be omitted. References are relevant and related to the subject but I would suggest adding more recent publications.

Response 5: We removed the double numeration of the reference list. We used the most recent publications in our analysis for review and we discussed this analysis with former studies. One reviewer recommended us even older studies.
